# CrowdGO: Machine learning and semantic similarity guided consensus Gene Ontology annotation

**Maarten J. M. F. Reijnders** [ID]*, **Robert M. Waterhouse** [ID]*

Department of Ecology and Evolution, University of Lausanne, and Swiss Institute of Bioinformatics, Lausanne, Switzerland

* mreijnders@live.nl (MJMFR); robert.waterhouse@unil.ch (RMW)

## Abstract

Characterising gene function for the ever-increasing number and diversity of species with annotated genomes relies almost entirely on computational prediction methods. These software are also numerous and diverse, each with different strengths and weaknesses as revealed through community benchmarking efforts. Meta-predictors that assess consensus and conflict from individual algorithms should deliver enhanced functional annotations. To exploit the benefits of meta-approaches, we developed CrowdGO, an open-source consensus-based Gene Ontology (GO) term meta-predictor that employs machine learning models with GO term semantic similarities and information contents. By re-evaluating each gene-term annotation, a consensus dataset is produced with high-scoring confident annotations and low-scoring rejected annotations. Applying CrowdGO to results from a deep learning-based, a sequence similarity-based, and two protein domain-based methods, delivers consensus annotations with improved precision and recall. Furthermore, using standard evaluation measures CrowdGO performance matches that of the community's best performing individual methods. CrowdGO therefore offers a model-informed approach to leverage strengths of individual predictors and produce comprehensive and accurate gene functional annotations.

## Author summary

New technologies mean that we are able to read the genetic blueprints in the form of complete genome sequences from many different species. We are also able to use computational methods combined with evidence from experiments to map out the locations in the genomes of many thousands of genes and other important regions. However, discovering and characterising the biological functions of all these genes and their protein products requires considerably more experimental work. In order to gain insights into the possible functions of the many genes currently lacking functional information from experiments we must therefore rely on methods that computationally predict protein functions. Many different software tools have been developed to tackle this challenge, each with their own strengths and weaknesses as shown by several community-based competitions that assess

**Data Availability Statement:** The source code and data used to produce the results and analyses presented in this manuscript are available from GitLab repository: https://gitlab.com/mreijnders/CrowdGO.

**Funding:** This work was supported by Swiss National Science Foundation (https://www.snf.ch/en) grants PP00P3_170664 and PP00P3_202669 to RMW. The funders had no role in study design, data collection and analysis, decision to publish, or preparation of the manuscript.

**Competing interests:** The authors have declared that no competing interests exist.

the performance of the predictors. Taking advantage of powerful modern machine learning techniques, we developed CrowdGO, a new software that aims to combine predictions from several tools and produce comprehensive and accurate gene functional annotations. CrowdGO is able to computationally assess agreements and conflicts amongst annotations from different predictors to then re-evaluate the results and deliver enhanced predictions of protein functions.

This is a *PLOS Computational Biology* Software paper.

## Introduction

New technologies and decreasing costs are enabling rapid accumulation of large quantities of genomic data. Experimental elucidation of biological functions of genomic elements such as protein-coding genes lags behind because it requires considerable additional efforts. Exploiting the potential of genomic data therefore relies on bioinformatics approaches to predict functions. The most comprehensive and widely-used model of describing gene function is the Gene Ontology (GO), a cornerstone for biology research [1,2]. Numerous computational tools have therefore been developed aiming to transfer functional information in the form of GO term annotations from functionally characterised macromolecules to those currently lacking assigned functions [3,4]. Many software focus on using protein sequences to predict function through sequence homology and/or domains and motifs [5–8]. Others use protein sequence features in combination with similarity based predictions, or make use of complementary data sources where available, such as protein structures or protein-protein interactions [9–11]. The function prediction community responded to the proliferation of tools with the Critical Assessment of Functional Annotation (CAFA) initiative to evaluate and improve computational annotations of protein function [12]. The common platform to benchmark performance reveals different strengths and weaknesses of participating predictors using standardised metrics. These measures are used to assess results from participating software for the Molecular Function Ontology (MFO), the Biological Process Ontology (BPO), and the Cellular Component Ontology (CCO).

Leveraging the strengths of individual algorithms through a consensus-conflict assessing meta-predictor should offer a means to achieve improved annotations of protein function. To test this potential, we developed CrowdGO, a consensus-based GO term meta-predictor that employs machine learning models with GO term semantic similarities and information contents (IC) to produce enhanced functional annotations. By re-evaluating each gene-term annotation, a consensus dataset is produced with high-scoring confident annotations and low-scoring rejected annotations. We assess the performance of CrowdGO results compared with four input annotation sets from a deep learning-based, a sequence similarity-based, and two protein domain-based methods. We also examine the effects of CrowdGO on reclassifying true and false positive and negative annotations. Using standard evaluation measures with the community benchmarking datasets, we compare CrowdGO results with those from CAFA3. Finally, we compare existing GO term annotations for several model and non-model species with those predicted using CrowdGO. The assessments and comparisons show that CrowdGO offers a model-informed approach to leverage strengths of input predictors and produce comprehensive and accurate gene functional annotations.

## Design and implementation

### The CrowdGO algorithm

CrowdGO examines similarities between annotations from different GO term predictors, using ICs, semantic similarities, and a machine learning model to re-evaluate the input annotations and produce consensus results. Input for CrowdGO consists of (i) gene-term annotations from individual GO term predictors; (ii) the GO Annotation (GOA) database for UniProt [13] proteins; and (iii) a pre-trained Adaptive Boosting (AdaBoost) [14] machine learning model. Annotations must be provided as a text file in tabular format consisting of four columns: predictor name, protein identifier, GO term identifier, and probability score. The GOA is provided with CrowdGO or can be downloaded from UniProt. Pre-trained AdaBoost models are provided, and new models can be trained using CrowdGO accessory functions.

The IC for each GO term from the input datasets is computed as the relative frequency of a GO term $i$ compared to the total number of GO terms in the UniProt GOA database:

$$IC(GO_i) = -log\left(\frac{\{go : go_i \in GOA\}}{\{go : go \in GOA\}}\right) \qquad (1)$$

The ICs are then used to compute semantic similarities across all pairs of GO terms assigned to each protein by any of the input methods. GO term semantic similarities employ the GO's directed acyclic graph (DAG) to define a metric representing the closeness in meaning between pairs of terms [15]. It is computed here with $S$ being the subset of GO terms shared between terms $i$ and $j$ after propagating up the GO DAG using the 'is_a' and 'part_of' relations, using the formula proposed by Lin [16]:

$$Sim\left(GO_i, GO_j\right) = \frac{2 \cdot max_{go \in S(go_i, go_j)}\{IC(go)\}}{IC(go_i) + IC(go_j)} \qquad (2)$$

Gene-term annotations from all input methods are then compared using these similarity scores. For each input dataset, support for a given annotation is collected in the form of a list of the most similar GO term from each of the other methods, requiring a semantic similarity score of 0.5 or higher. Each gene-term annotation is therefore associated with the provided predictor method and probability score, and the GO term's IC. When other methods predict the same or similar GO terms then the support list for each annotation comprises each of these additional GO terms, the provided predictor method and probability score, the computed IC, and the semantic similarity score with the GO term of the annotation they support. The support lists for each annotation are then used as features in pre-trained AdaBoost machine learning models. Based on these features, the model returns a probability score for each unique annotation from all input datasets. These are used to re-evaluate each annotation and produce the consensus dataset split into confident annotations with scores of 0.5 or higher, and the remaining rejected annotations. The threshold of 0.5 is the default value from AdaBoost applied here as the standard threshold for coherence across the three ontologies, which in each case closely match the threshold values that correspond to their $F_{max}$ scores (0.510 for MFO, 0.477 for BPO, and 0.495 for CCO).

CrowdGO provides four pre-trained models in pickle serialized file format: CrowdGOFull, CrowdGOLight, CrowdGOFull-SwissProt, and CrowdGOLight-SwissProt. CrowdGOFull is trained using results from four predictors: DeepGOPlus [11] Wei2GO [17]; InterProScan [18]; and FunFams [19]. These are open access tools with annotation database versions that can be selected to match exactly the date requirements for building annotation datasets that do not overlap in content and can thus be used for comparing performances. Of the provided models,

CrowdGOLight excludes InterProScan, and the '-SwissProt' models use input from Wei2GO based only on SwissProt proteins rather than the full UniProt set. The pre-trained AdaBoost models are built by supervised learning with labelled training data consisting of annotations from SwissProt (see S1 Text). The CrowdGOFull model requires input from all four predictor methods, and uses the TrEMBL protein database. The other models reduce the computations by excluding InterProScan (Light models) and the reference dataset sizes by requiring only the smaller SwissProt protein database. The scikit-learn [20] AdaBoost models are trained using a decision tree classifier with a depth of 2 as base estimator, a maximum of 100,000 iterations, a learning rate of 1, and the SAMME algorithm. AdaBoost scores are calibrated using the scikit-learn CalibratedClassifierCV package with 10-fold cross validation. Importantly, users may train their own new models incorporating additional predictors using the accessory functions provided with CrowdGO.

Details of the proteins and their gene ontology annotations used for building the training, testing, and CAFA3 benchmarking sets are provided in S1 Text. These also describe the implementations required for performing the assessments, including ensuring non-overlapping testing and training sets and software versions, definitions of the evaluation metrics used, steps to enumerate correct and incorrect classifications of true and false positive and negative annotations, workflows for the annotations of the proteomes of model and non-model species, and a step-by-step guide to running CrowdGO.

## Results

### Training, testing, and benchmarking annotations

To assess the performance of CrowdGO we first built training, testing, and benchmarking GO annotation sets following best practices for evaluating machine learning models (Table A in S1 Text) and CAFA3 benchmark generation guidelines (see S1 Text). The set of annotations for building the CrowdGO models was derived from all proteins added to the GO Annotation (GOA) UniProt database from 2018 (version 162) to 2020 (version 198). The CAFA3 benchmarking dataset is older and consists of new GO term annotations added to the GOA UniProt database during 2017 as detailed in [12]. The V162-V198 training and testing annotation datasets have well-matched numbers of proteins per ontology, and a little over two thirds as many for MFO, about 10% more for BPO, and about double for CCO when compared with the CAFA3 benchmarking dataset (Table 1). The average numbers of leaf and parent-propagated GO terms per protein are generally similar across the three datasets, apart from slightly higher averages for CAFA3 MFO and nearly double the leaf annotations per protein for BPO in the V162-V198 datasets. Terms per protein and term ICs are also well-matched between the training and testing datasets, showing no significant differences in their distributions (Figs A and B in S1 Text). The sequence identity distribution of the V162-V198 training and testing datasets follows a pattern very similar to those shown previously for the CAFA datasets (Fig C in S1 Text), therefore no sequence-similarity-based redundancy-reducing was performed. In the context of building a machine learning model, the V162-V198 dataset therefore offers a representative baseline to assess the performance of CrowdGO consensus results compared with annotations from individual input predictors. The CAFA3 benchmarking dataset is the standard in the field of protein function prediction, and thus offers a community reference to assess the performance of CrowdGO results compared with annotations from top-performing CAFA3 predictors.

### CrowdGO consensus compared with input predictors

Next we used the V162-V198 training and testing annotation datasets to assess the performance of CrowdGO compared with annotations from four different individual input

**Table 1. Average GO terms per protein for the V162-V198 training and testing datasets and the CAFA3 benchmarking dataset.** Molecular Function (MFO), Biological Process (BPO), and Cellular Component (CCO) ontologies; leaves: GO terms with no child terms annotated for the same protein; propagated: after propagating parent GO term annotations.

| Dataset | Proteins | MFO | | BPO | | CCO | |
|---|---|---|---|---|---|---|---|
| | | leaves | propagated | leaves | propagated | leaves | propagated |
| **V162-V198 training** | 335 MFO<br>988 BPO<br>1244 CCO | 1.3 | 7.4 | 6.8 | 32.0 | 4.7 | 13.7 |
| **V162-V198 testing** | 326 MFO<br>1000 BPO<br>1222 CCO | 1.3 | 7.3 | 6.1 | 29.4 | 4.6 | 13.5 |
| **CAFA3 benchmarking** | 454 MFO<br>848 BPO<br>639 CCO | 1.7 | 9.2 | 3.6 | 29.0 | 4.7 | 13.8 |

predictors comprising a deep learning-based, a sequence similarity-based, and two protein domain-based methods. Importantly, all datasets used to inform these predictors predate the annotated sequences in the V162-V198 dataset to ensure no overlaps, and the CrowdGO model was built with results from annotating the V162-V198 training dataset using these same predictors and datasets (see S1 Text). Precision-recall curves show the performance of individual results from DeepGOPlus [11], FunFams [19], InterProScan [18], and Wei2GO [17] compared with results from applying CrowdGO with the CrowdGOFull model to predictions from all four methods (Fig 1). For each of the ontologies, CrowdGO results show increased precision over annotations from the individual input predictors. Furthermore, CrowdGO presents a much smoother relationship between precision and recall across thresholds: the highest scoring annotations are the most reliable, and the lower scoring annotations steadily increase recall while reducing reliability. Evaluation metrics computed from the precision-recall curves (see S1 Text) show performance in terms of $F_{max}$, the maximum protein-centric F-measure, $S_{min}$ the Information Content (IC) semantic distance between true and predicted annotations, and area under the precision-recall curve (AUPR). CrowdGO results with the CrowdGOFull model show improved $F_{max}$, $S_{min}$, and AUPR scores compared with annotations from each input predictor for each ontology (Table 2). The largest improvements across all evaluation metrics are with respect to the two protein domain-based methods, FunFams and InterProScan. The $F_{max}$ improvements (Fig 1D) are largest for MFO, where the two next-best methods score similarly (DeepGOPlus and Wei2GO), but smaller for BPO and CCO where each has only one similarly-scoring next-best method. Compared to the CrowdGOFull model, the 'Light' and 'SwissProt-only' models show overall lower $F_{max}$ and AUPR scores with higher $S_{min}$ values, while still performing better than the individual input predictors by almost all measures. Assessing performance using the V162-V198 annotation dataset therefore demonstrates that the consensus approach implemented by CrowdGO successfully improves precision, recall, and $S_{min}$ of gene-term annotations from each of the four individual input predictors.

## Impact of CrowdGO annotation re-evaluations

Because CrowdGO performs model-informed re-evaluations of scores from different input predictors, we next sought to quantify the impact of re-evaluations on the resulting annotations. Consensus predictors aim to leverage the information from multiple input methods to refine scoring and thereby improve overall calling of true positives and true negatives while decreasing false positives and false negatives. Thresholds based on the $F_{max}$ achieved by each

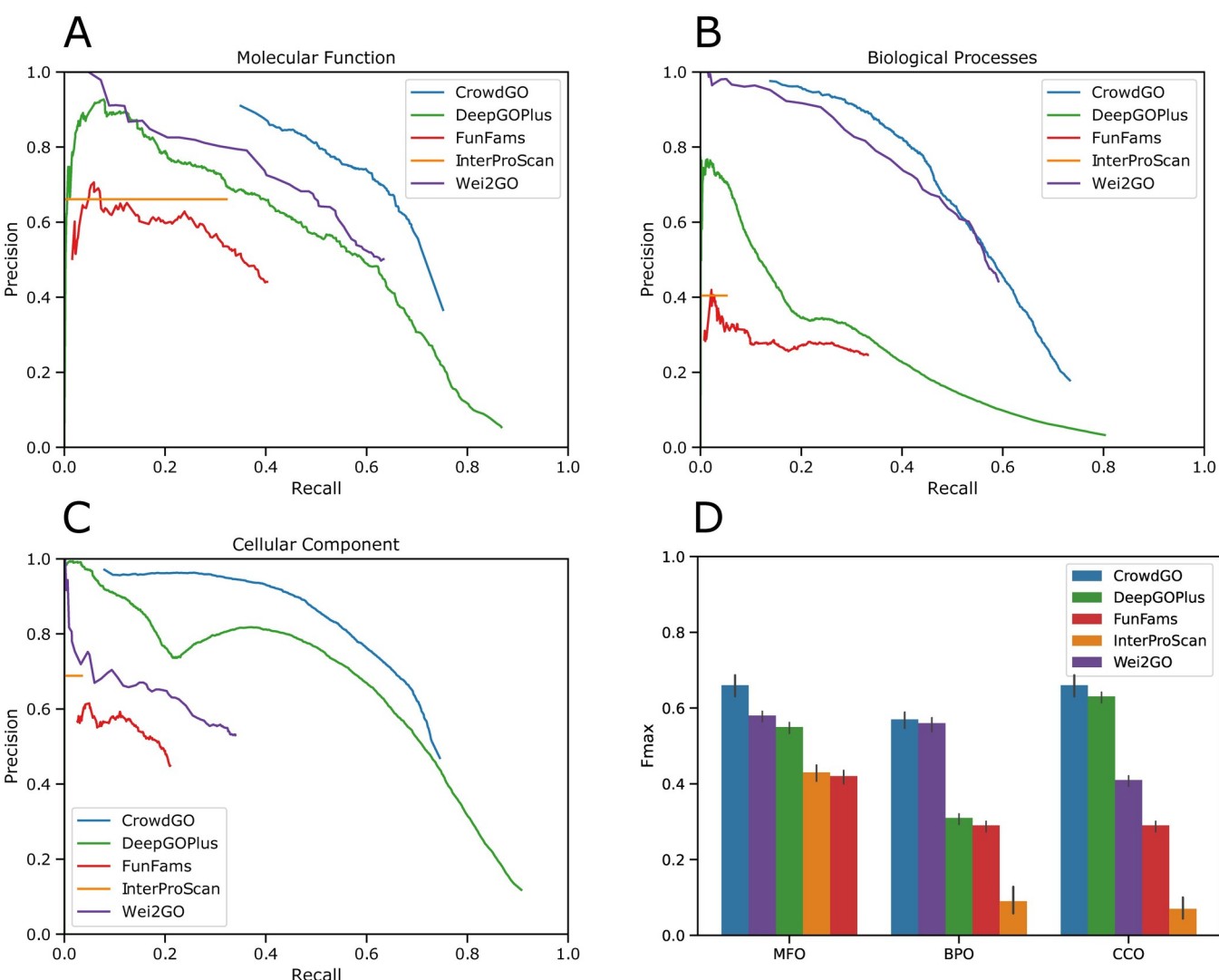

**Fig 1. Precision-recall curves for GO term annotations from CrowdGO and four individual input predictors.** Performance was assessed using the CrowdGOFull model on the V162-V198 annotation dataset with threshold steps of probability scores of 0.01 for each predictor (except for InterProScan) for (A) the Molecular Function Ontology, (B) the Biological Process Ontology, and (C) the Cellular Component Ontology. $F_{max}$ scores are shown in (D), sorted from highest to lowest scoring method for each ontology, with bars showing 95% confidence intervals calculated using 10,000 bootstrap iterations. Root terms are excluded from the evaluation. InterProScan predictions all have the same prediction probability (set to one), precision-recall curves are therefore presented as horizontal lines from zero recall to the actual recall.

method using the V162-V198 annotation dataset were used to label true and false positives and negatives of the input annotations before applying CrowdGO (with the CrowdGOFull model) and assessing the impacts of the re-evaluations (see S1 Text). By far the largest change for each of the three ontologies is the correction of false positives to true negatives (Fig 2). These changes dwarf the inverse, incorrect re-evaluations by CrowdGO that convert true negatives to false positives: correct re-evaluations are 11–56 times more numerous. The vast majority of true positives (75%-89%) and true negatives (84%-98%) are correctly affirmed, i.e. not re-classified, after applying CrowdGO. The consensus results are also able to recover false negatives from the input predictors and correct them to true positives, albeit for much smaller subsets. However, the consensus is somewhat conservative and therefore also incorrectly converts some true positives into false negatives. Comparing counts of true and false positives,

**Table 2. Evaluations of GO term annotations from CrowdGO and four individual input predictors.** Molecular Function (MFO), Biological Process (BPO), and Cellular Component (CCO) ontologies; Highest $F_{max}$ scores and AUPR values, and lowest $S_{min}$ scores are in bold text.

| Method | $F_{max}$ | | | $S_{min}$ | | | AUPR | | |
|---|---|---|---|---|---|---|---|---|---|
| | MFO | BPO | CCO | MFO | BPO | CCO | MFO | BPO | CCO |
| **CrowdGOFull** | **0.65** | **0.58** | **0.68** | **7.75** | **43.45** | **16.60** | **0.59** | **0.54** | **0.67** |
| **CrowdGOLight** | 0.63 | **0.58** | **0.68** | 8.12 | 43.52 | 16.68 | 0.56 | **0.54** | **0.67** |
| **CrowdGOFull-SwissProt** | 0.63 | 0.56 | 0.66 | 8.25 | 45.80 | 17.24 | 0.56 | 0.53 | 0.65 |
| **CrowdGOLight-SwissProt** | 0.61 | 0.55 | 0.65 | 9.03 | 46.17 | 17.99 | 0.55 | 0.51 | 0.64 |
| **DeepGOPlus** | 0.55 | 0.31 | 0.63 | 11.39 | 86.19 | 19.96 | 0.50 | 0.22 | 0.61 |
| **FunFams** | 0.42 | 0.29 | 0.29 | 11.52 | 87.47 | 23.23 | 0.23 | 0.09 | 0.10 |
| **InterProScan** | 0.43 | 0.09 | 0.07 | 12.70 | 88.74 | 24.63 | 0.21 | 0.02 | 0.02 |
| **Wei2GO** | 0.58 | 0.56 | 0.41 | 8.53 | 47.55 | 20.81 | 0.49 | 0.48 | 0.22 |

CrowdGO consistently maximises true positive calls while minimising false positives compared to the results from the four different input predictors individually (Fig 2). The ratios are thus consistently improved over the individual inputs, and in several cases considerably so, e.g. for BPO from four false positives for each true positive for DeepGOPlus, and eight false positives for each true positive for FunFams, to just one false positive for every three true positives. Assessing gene-term annotation re-evaluations using the V162-V198 annotation dataset therefore demonstrates that CrowdGO successfully decreases the calling of false positives while improving the overall calling of true positives and true negatives.

## CrowdGO comparisons with CAFA3 methods

We next used the CAFA3 benchmarking dataset as a reference to compare CrowdGO results with annotations from baseline methods and predictors that took part in CAFA3 [12]. The four provided CrowdGO models (i.e. pre-trained on the complete V162-V198 dataset) were applied to GO term annotations produced by the four input predictors on the CAFA3 benchmarking dataset (see S1 Text). Performance summarised using the $F_{max}$ scores (Table 3) shows that CrowdGO achieves $F_{max}$ scores equal to the top-performing CAFA3 method for BPO, slightly lower than the best-scoring method but higher than the next-best performing methods for MFO, and slightly higher than the best CAFA3 methods for CCO. By this evaluation, the consensus approach applied to results from just four input predictors is able to produce annotations with precision and recall matching the $F_{max}$ scores of the best-performing CAFA3 methods.

## CrowdGO applied to model and non-model species

Having established that CrowdGO delivers consensus annotations with improved precision and recall matching the community's best performing methods, we next assessed the ability of CrowdGO to functionally annotate complete proteomes of taxonomically diverse species. CrowdGO was applied to the full sets of protein-coding genes from a selection of animals, plants, fungi, and bacteria (see S1 Text). The confident consensus results were compared to existing GO term annotations from UniProt with respect to numbers of annotated proteins, and distributions of numbers of GO Slim terms per protein and total leaf+parents GO term informativeness (Fig 3, Fig D in S1 Text). Firstly, across the ontologies total numbers of annotated proteins increase for all 12 species, with larger increases for non-model species that have generally less comprehensive experimental data to support functional annotations. While such increased annotation coverage could be expected from simply applying individual homology-based methods, here CrowdGO achieves increased coverage while maximising true positives

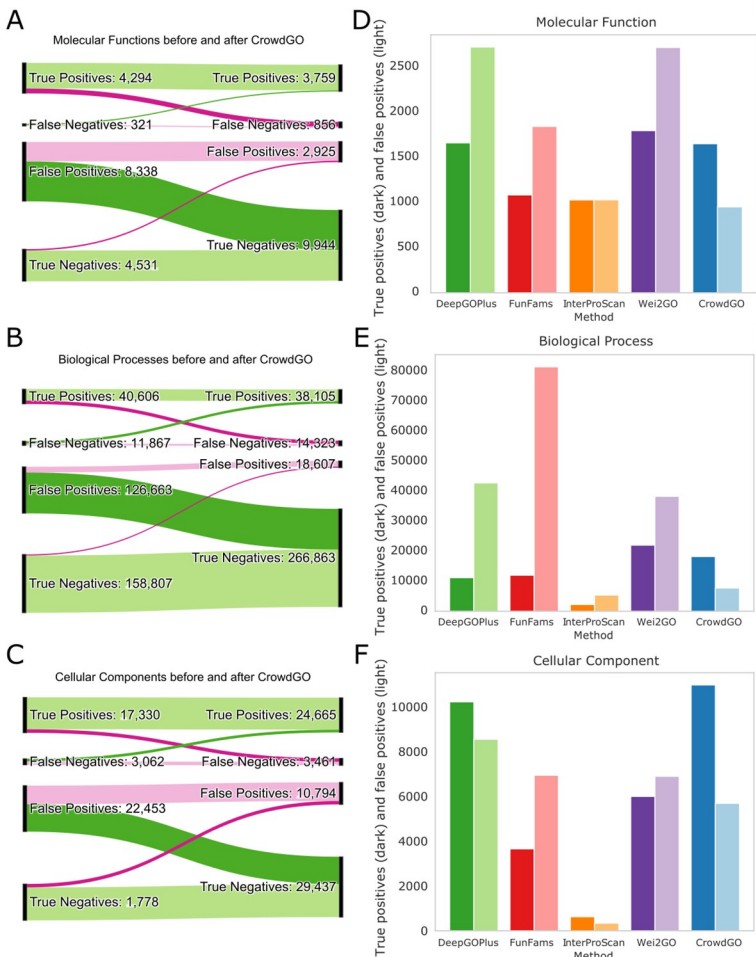

**Fig 2. Impact of CrowdGO annotation re-evaluations on calling true and false positives and negatives.** Sankey plots show the numbers of true or false positive or negative gene-term annotations before (left) and after (right) re-evaluation by CrowdGO (using the CrowdGOFull model) for the three gene ontologies, (A) Molecular Function, (B) Biological Process, and (C) Cellular Component, for the V162-V198 testing dataset. Correct re-evaluations are shown in dark green (false positives to true negatives and false negatives to true positives), and incorrect re-evaluations are shown in dark pink (true positives to false negatives and true negatives to false positives). For unchanged annotations, correct affirmations are shown in light green (true positives and true negatives), and incorrect affirmations are shown in light pink (false positives and false negatives). Bar plots show for each method and CrowdGO (coloured as in Fig 1) counts of true positives (dark colours) and false positives (light colours) for the three gene ontologies, (D) Molecular Function, (E) Biological Process, and (F) Cellular Component, for the V162-V198 testing dataset.

and minimising false positives (Fig 2). CrowdGO annotations overall outnumber those obtained using a baseline homology-based approach (but slightly fewer for BPO), with 53% MFO, 44% BPO, and 39% CCO of CrowdGO annotations also being predicted by the baseline approach (Table B in S1 Text). Secondly, the breadth of functional annotations also increases, with medians increasing by one to four GO Slim terms per protein compared to the TrEMBL or UniProt annotation datasets. Finally, the depth of functional annotations in terms of total informativeness of annotations per protein also increases, most notably for non-model species and compared to TrEMBL annotation datasets of model species. For model species with sub-sets of electronically inferred TrEMBL and manually reviewed SwissProt annotations such as *Arabidopsis thaliana* (Fig 3A) and *Drosophila melanogaster* (Fig 3B), the SwissProt annotations show more GO Slim terms and higher total informativeness per protein, albeit with fewer

**Table 3. $F_{max}$ scores for CrowdGO and the baseline and top three performers from CAFA3.** Molecular Function (MFO), Biological Process (BPO), and Cellular Component (CCO) ontologies; Highest $F_{max}$ scores are in bold text.

| Method | $F_{max}$ | | |
|---|---|---|---|
| | **MFO** | **BPO** | **CCO** |
| **CAFA3 #1 (Zhu Lab [21])** | **0.62** | **0.40** | 0.61 |
| **CAFA3 #2 (orengo-funfams [5], INGA-Tosatto [22], Kihara Lab [23])** | 0.54 | 0.39 | 0.61 |
| **CAFA3 #3 (Tian Lab [24], Argot25Toppo Lab [7], INGA-Tosatto [22])** | 0.53 | 0.38 | 0.60 |
| **CAFA3 range #4-#10 [12]** | 0.51–0.53 | 0.37–0.38 | 0.58–0.60 |
| **CAFA3 Naïve** | 0.33 | 0.26 | 0.54 |
| **CAFA3 BLAST** | 0.42 | 0.26 | 0.46 |
| **CrowdGOFull** | 0.57 | **0.40** | **0.62** |
| **CrowdGOLight** | 0.57 | **0.40** | **0.62** |
| **CrowdGOFull-SwissProt** | 0.57 | 0.38 | 0.61 |
| **CrowdGOLight-SwissProt** | 0.57 | 0.39 | 0.60 |

annotated proteins. In both these species, annotation breadth and depth of the high-quality SwissProt subsets more closely resemble those of CrowdGO than TrEMBL annotations, despite CrowdGO covering the largest numbers of proteins. Thus CrowdGO dramatically increases proteome coverage compared to SwissProt, while adding more terms, and more informative terms, compared to TrEMBL. The model bacterium, *Escherichia coli* (Fig 3C), and yeast, *Saccharomyces cerevisiae* (Fig 3D), are both fully SwissProt-reviewed, so CrowdGO only marginally increases proteome coverage. Compared with these manually curated gold standards, the automated predictions inevitably result in decreased annotation breadth and depth. The non-model species on the other hand have few or no SwissProt-reviewed annotations, thus their full UniProt annotations (TrEMBL+SwissProt) are used for comparisons, where CrowdGO results show improved proteome coverage with more GO Slim terms and higher total informativeness per protein. Notably, breadth and depth of CrowdGO annotations of the proteomes of these non-model species more closely resemble those of the high-quality SwissProt annotations for the model species than their own UniProt annotations.

## Availability and future directions

CrowdGO is available from https://gitlab.com/mreijnders/CrowdGO. The distribution includes DeepGOPlus (GitHub version 01-06-2020), FunFams (GitHub version 01-06-2020), and Wei2GO (GitLab version 15-11-2020). The user guide details instructions for also including InterProScan. Snakemake [25] workflows are provided to run each of the individual predictors, and to run CrowdGO to produce consensus annotations. CrowdGO can also be run directly with Python and the user-provided results from individual predictors, as detailed in the user guide. The distribution includes the four pre-trained AdaBoost models as well as supporting scripts to train new CrowdGO prediction models, e.g. when using other predictors to generate input annotations.

Here we explore whether a consensus-based GO term meta-predictor that leverages the strengths of individual algorithms can achieve improved computational annotations of protein function. Our results demonstrate that enhanced sets of functional annotations can be produced by employing machine learning models with GO term semantic similarities and ICs to re-evaluate sets of input annotations. Assessing CrowdGO consensus annotations using a dataset derived from all new SwissProt proteins over a 2.5-year period shows improved precision, recall, and $S_{min}$ over each of four individual input predictors. Examining the effects of annotation re-evaluations further demonstrates that CrowdGO successfully decreases the calling of

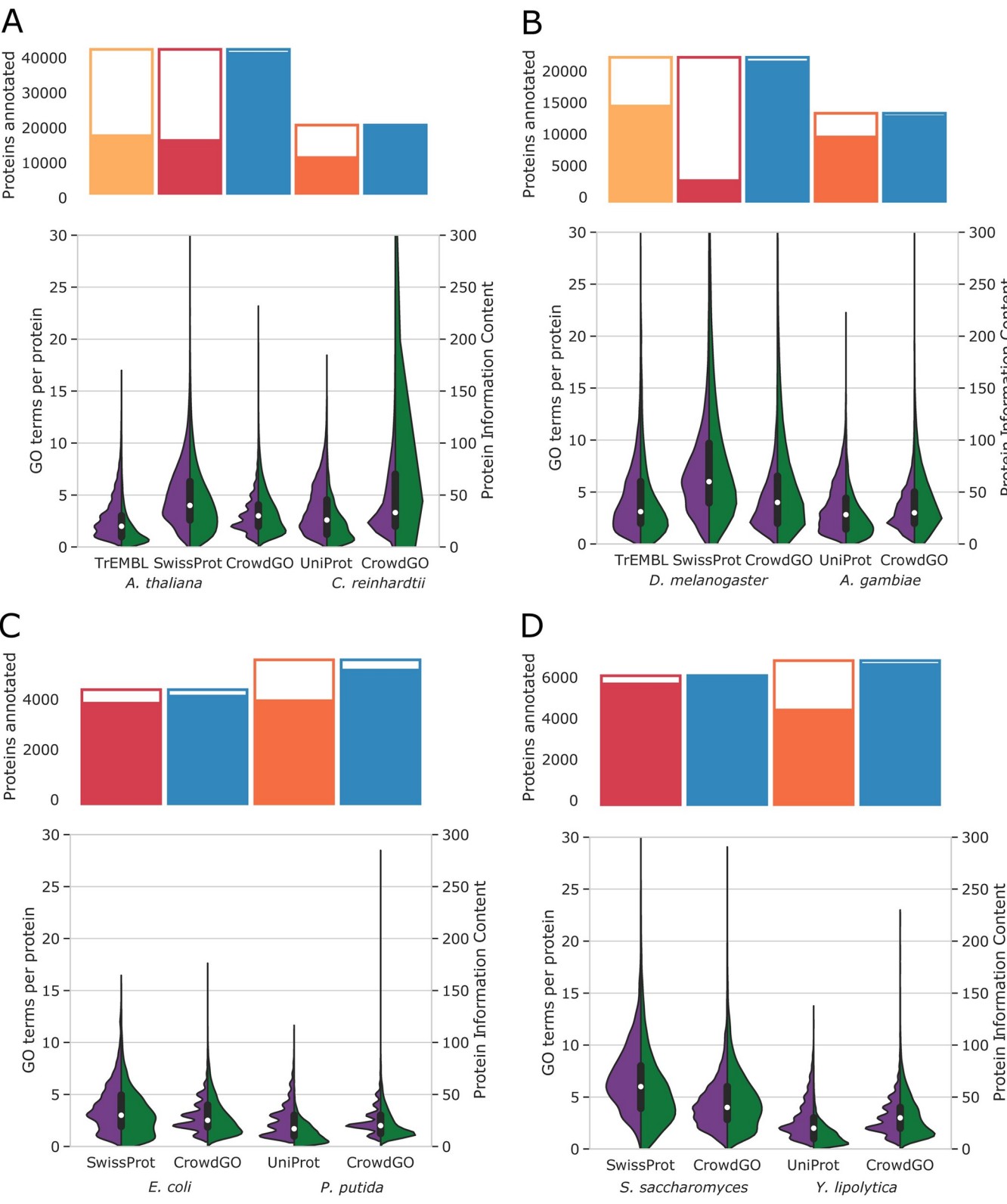

**Fig 3. Comparisons of whole-proteome CrowdGO annotations and existing UniProt annotations.** CrowdGO consensus annotation results (using the CrowdGOFull model) were compared with existing Gene Ontology (GO) term annotations from UniProt, and with the subsets of manually curated SwissProt (where available) and automatically inferred TrEMBL annotations, for representative taxa of (A) plants and algae, (B) insects, (C) bacteria, and (D) fungi. For each species traditionally considered as a model species (*A. thaliana*, *D. melanogaster*, *E. coli*, and *S. cerevisiae*) a non-model species is used for comparison (*C.*

*reinhardtii*, *A. gambiae*, *P. putida*, and *Y. lipolytica*). Bars at the top of each panel show the total numbers of proteins annotated with at least one GO term for each respective annotation dataset, with white-filled areas showing the remaining proteins with no annotations. The most general annotations are excluded from the total numbers of proteins annotated by counting only GO terms with at least two parents. Split violin plots show the distributions of the numbers of GO Slim terms annotated per protein (purple, left), and summed leaf+parents information content (IC) per protein (green, right). Y-axes are limited to a maximum of value of 300 for total protein IC distributions. The boxplots show the medians and 1.5 times the interquartile range for the numbers of GO Slim terms annotated per protein.

false positives while improving the overall calling of true positives and true negatives. Using the CAFA3 benchmarking dataset shows that applying CrowdGO to results from just four input predictors is able to produce annotations with precision and recall matching the $F_{max}$ scores of the best-performing CAFA3 methods. Annotating complete proteomes of a diverse selection of species demonstrates that, while consensus predictions cannot match gold standard models like *E. coli* and *S. cerevisiae*, they can substantially improve coverage, breadth, and depth of functional annotations for a wide range of organisms. Using a model-informed meta-predictor approach, CrowdGO is able to re-evaluate results from individual predictors and produce comprehensive and accurate functional annotations.

We employed as input for model training just four predictors: the deep learning-based DeepGOPlus method, the sequence similarity-based Wei2GO approach, as well as the InterProScan and FunFams protein domain-based methods. DeepGOPlus and Wei2GO were selected because they are both entirely open-source and easily implementable. FunFams is the only method of these four that participated in CAFA3, where it performed well (amongst the top 10 for all three ontologies). It is also open-source, although to obtain annotation scores we had to implement scoring based on details provided in [5]. InterProScan is freely available and widely used for annotating protein domains, with quality-assured InterPro2GO domain-to-term mapping generated through manual curation [26]. We did not include the overall CAFA3 best performer, GOLabeler [21], because without an available open-source implementation it is not possible to train the machine learning classifiers that CrowdGO relies on to perform the consensus re-evaluations. Other CAFA3 top-performers offer only in-house or web-server implementations, and thus currently cannot be used with CrowdGO. Future open-source implementations of these and other newer methods would allow for their integration into CrowdGO through providing additional sets of pre-trained models.

Our evaluations indicate that CrowdGO performance in terms of $F_{max}$ scores is primarily driven by one or two of the input predictors, namely DeepGOPlus for CCO, Wei2GO for BPO, and both these methods for MFO (Fig 1D). The two domain-based methods contribute less overall, particularly for CCO and BPO predictions, but with more substantial contributions for MFO, where CrowdGO improves the most over the next-best methods. A simple consensus approach such as retaining only annotations from the best-performing input method that are also supported by at least one other method could still reject true positives. Instead, the CrowdGO consensus approach performs model-informed re-evaluations of the scores of all annotations based on predictions from all inputs, enabling it to improve on the best input method, especially for MFO. Therefore, assuming that good models can be built with additional methods, the inclusion of more input annotation sets could lead to enhanced performance improvements. Furthermore, the re-evaluations also produce scores that more smoothly balance precision and recall allowing users to select larger sets of less confident annotations or to focus on only the highest scoring most reliable annotations.

The CrowdGO models were built using an Adaptive Boosting [14] machine learning model, which aims to combine a set of weak classifiers into a weighted sum representing the boosted strong classifier. Future implementations of CrowdGO, especially with the inclusion of additional input predictors, might benefit from developing additional models using

eXtreme Gradient Boosting, a popular alternative that is a powerful algorithm for supervised learning tasks [27]. Given the variety of formulations used for measuring semantic similarities [15,28], and the recognition of the power of using ontologies to compute similarity and incorporate them in machine learning methods [29], future developments to CrowdGO could also offer users models that are based on alternatives to the currently implemented Lin's measure. Results from applying the model-informed approach of CrowdGO demonstrate that a consensus meta-predictor can improve protein functional annotations, as well as presenting opportunities for future models built with new training data, incorporating new predictors, and alternative boosting, to further enhance performance.

## Supporting information

**S1 Text. Supplementary Materials and Methods.** Table A: Checklist for reporting and evaluating machine learning models. Table B: Comparisons of CrowdGO annotations with baseline homology-based annotations. Fig A: Distributions of GO terms per protein for the V162-V198 training and testing datasets and the CAFA3 benchmarking dataset. Distributions are shown as density plots (A-C) and as empirical cumulative distribution function (ECDF) plots (D-E) for the Molecular Function (MFO), Biological Process (BPO), and Cellular Component (CCO) ontologies. The total number of proteins in each dataset are shown on the plots, along with results from Wilcoxon (Mann-Whitney) tests for each pair of datasets for A-C, and with results of Kolmogorov-Smirnov tests for each pair of datasets for D-F. Analysis, plotting, and significance testing all performed using R 3.6.1. The V162-V198 training and testing datasets show no significant differences in their distributions of GO terms per protein. Compared with the CAFA3 dataset, they are both significantly lower for MFO, significantly higher for BPO, and not significantly different for CCO. Fig B: Distributions of information contents for GO terms from the V162-V198 training and testing datasets and the CAFA3 benchmarking dataset. Distributions are shown as density plots (A-C) and as empirical cumulative distribution function (ECDF) plots (D-E) for the Molecular Function (MFO), Biological Process (BPO), and Cellular Component (CCO) ontologies. The total number of gene-term annotations in each dataset are shown on the plots, along with results from Wilcoxon (Mann-Whitney) tests for each pair of datasets for A-C, and with results of Kolmogorov-Smirnov tests for each pair of datasets for D-F. Analysis, plotting, and significance testing all performed using R 3.6.1. The V162-V198 training and testing datasets show no significant differences in their distributions of information contents (ICs). Compared with the CAFA3 dataset, their medians are not significantly different for MFO but their distributions are significantly flatter (the peak in the CAFA3 dataset arises from GO:0042802 'identical protein binding' with an IC of 6.5 annotated to 61 proteins, the most proteins for any term). Their ICs are significantly higher than the CAFA3 dataset for both BPO and CCO. Fig C: Sequence identity distributions of the V162-V198 training and testing datasets. The sequence identity distribution of the V162-V198 training and testing datasets is shown as a histogram with proportions calculated as the number of amino acid matches divided by sequence length of the query protein based on the top-scoring hit from BLASTp searches with default settings. Fig D: CrowdGO applied to functionally annotate the complete proteomes of taxonomically diverse species. Results for eight species are shown in main text Fig 3, here results for the remaining four species are presented. CrowdGO consensus annotation results were compared with existing Gene Ontology (GO) term annotations from UniProt, and with the subsets of manually curated SwissProt (where available) and automatically inferred TrEMBL annotations, for (A) Pan troglodytes Chimpanzee and Homo sapiens human, (B) Arabidopsis thaliana the thale cress (repeated here for plant-plant model-non-model comparison) and Solanum lycopersicum Tomato, and (C) Candidatus Thorarchaeota archaeon SMTZ1-45.

Bars at the top of each panel show the total numbers of proteins annotated with at least one GO term for each respective annotation dataset, with white-filled areas showing the remaining proteins with no annotations. Split violin plots show the distributions of the numbers of GO Slim terms annotated per protein (purple, left), and summed leaf+parents information content (IC) per protein (green, right). Y-axes are limited to a maximum of value of 300 for total protein IC distributions. The boxplots show the medians and 1.5 times the interquartile range for the numbers of GO Slim terms annotated per protein.
(DOCX)

## Acknowledgments

The authors thank Romain Feron, Livio Ruzzante, and Antonin Thiébaut for providing useful suggestions for improvements and valuable feedback on the manuscript.

## Author Contributions

**Conceptualization:** Maarten J. M. F. Reijnders, Robert M. Waterhouse.

**Formal analysis:** Maarten J. M. F. Reijnders.

**Funding acquisition:** Robert M. Waterhouse.

**Investigation:** Maarten J. M. F. Reijnders, Robert M. Waterhouse.

**Methodology:** Maarten J. M. F. Reijnders, Robert M. Waterhouse.

**Project administration:** Maarten J. M. F. Reijnders, Robert M. Waterhouse.

**Software:** Maarten J. M. F. Reijnders.

**Supervision:** Robert M. Waterhouse.

**Validation:** Maarten J. M. F. Reijnders, Robert M. Waterhouse.

**Visualization:** Maarten J. M. F. Reijnders, Robert M. Waterhouse.

**Writing – original draft:** Maarten J. M. F. Reijnders, Robert M. Waterhouse.

**Writing – review & editing:** Maarten J. M. F. Reijnders, Robert M. Waterhouse.

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
