## [Decision Letter · Decision Letter 0]

25 Oct 2021

Dear Dr. Waterhouse,

Thank you very much for submitting your manuscript "CrowdGO: machine learning and semantic similarity guided consensus Gene Ontology annotation" for consideration at PLOS Computational Biology.

As with all papers reviewed by the journal, your manuscript was reviewed by members of the editorial board and by two independent reviewers. In light of the reviews (below this email), we would like to invite the resubmission of a significantly-revised version that takes into account the reviewers' comments.  Please pay particular attention to the major comments of Reviewer 1.

We cannot make any decision about publication until we have seen the revised manuscript and your response to the reviewers' comments. Your revised manuscript is also likely to be sent to reviewers for further evaluation.

Sincerely,

Jacquelyn S. Fetrow

Associate Editor

PLOS Computational Biology

Dina Schneidman

Software Editor

PLOS Computational Biology

Reviewer's Responses to Questions

**Comments to the Authors:**

Reviewer #1: The review is uploaded as an attachment.

Reviewer #2: Summary/General Remarks

The manuscript describes a software tool that combines predicted functional annotations from different GO predictors using an ensemble method leading to improved performance. The authors benchmark their tool against the top-performing method from CAFA3 and report competitive performance for biological process and cellular component and slightly worse performance for molecular function predictions.

The code is available with an open source license and contains a detailed README file with instructions on how to install and run the software and its dependencies. Currently, a few GO predictors are implemented and integrated, but it is straightforward for users to include more predictors. The tool could be of use to biologists who are interested in annotating new sequences as well as protein function prediction researchers as it provides a relatively strong baseline to compare new methods against.

Specific comments to the authors

1. Line 100: The input consists of predicted GO annotations from different methods in the form of posterior probabilities. Do these probabilities need to be consistent with the GO DAG (i.e. P(ancestor) >= P(descendant) )? Or does crowdGO internally make this correction?

2. Similar question for the output of crowdGO: Are the predictions guaranteed to respect the DAG hierarchy and if not, are they post-processed to make them so?

3. If someone wants to incorporate an additional predictor to a trained crowdGO model, does that imply that the adaboost algorithm has to be retrained from scratch?

4. Could the authors comment on the training time of the adaboost algorithm and its relation to the number of predictors used? Does it take considerably longer to train a model that combines more predictors?

5. The currently implemented predictors are all sequence-based, but for some model species, one might want to incorporate other data sources such as gene expression or protein interaction data, which are not available for most non-model species. Does crowdGO support “missing predictors” or would I need a separately trained model?

**Have the authors made all data and (if applicable) computational code underlying the findings in their manuscript fully available?**

Reviewer #1: Yes

Reviewer #2: Yes

PLOS authors have the option to publish the peer review history of their article (what does this mean?). If published, this will include your full peer review and any attached files.

Reviewer #1: No

Reviewer #2: No
---

## [Decision Letter · Decision Letter 1]

14 Feb 2022

Dear Dr. Waterhouse,

Thank you very much for submitting your manuscript "CrowdGO: machine learning and semantic similarity guided consensus Gene Ontology annotation" for consideration at PLOS Computational Biology. As with all papers reviewed by the journal, your manuscript was reviewed by members of the editorial board and by several independent reviewers. The reviewers appreciated the attention to an important topic. Based on the reviews, we are likely to accept this manuscript for publication, providing that you modify the manuscript as follows.  

Reviewer 1 continues to have comments on the revised manuscripts.  You have responded adequately to Reviewer 1's comments, with one exception which we'd like to determine if you can better address.  

**The original Reviewer 1 request and your response to this request was:**

**5. Application to full proteomes**: The authors state that CrowdGO increases the number of annotated proteins and the number of GO terms annotated to a certain protein. However, how many of those annotations could already have been obtained by e.g., using homology-based inference?

**Response:** Here we aim to show that by applying CrowdGO to annotating full proteomes users can obtain annotation sets with increased numbers of annotated proteins and GO terms annotated to a certain protein compared to simply sourcing existing annotations from SwissProt, UniProt, or TrEMBL. It is true that many of these annotations could also be obtained by applying homology-based inference methods, but as shown in Figure 2 such methods can produce many false positives. Therefore the advantage of the CrowdGO consensus approach is that it results in more confident annotations without reducing the overall coverage of annotations compared with those that can be sourced from standard community databases. We have added this important aspect to the text to clarify this for the readers.

**Reviewer 1 makes the following request based on the revised manuscript:**

**4. Application to full proteomes**: I agree that just applying homology-based inference without any threshold will result in many false positive. However, how many of the annotations from CrowdGO could be obtained through homology-based inference using a common E-value threshold or similar? Again, I understand why applying CrowdGO is (probably) superior to applying homology-based inference. However, if the authors believe that why not support it with the respective results?

Please consider if CrowdGO's improvement can actually be quantified to demonstrate clearly how much better that CrowdGO was from simple homology modeling.  Otherwise, the manuscript is acceptable and does not need to respond to Reviewer 1's remaining comments on this revision.

Sincerely,

Jacquelyn S. Fetrow

Associate Editor

PLOS Computational Biology

Dina Schneidman

Software Editor

PLOS Computational Biology

[LINK]

Reviewer's Responses to Questions

**Comments to the Authors:**

Reviewer #1: Most of my previous comments were addressed satisfactorily by the authors. However, for many, the authors only provide arguments against the proposed analyses or speculations of their results. While I understand that performing additional analyses can be time-consuming, I still believe that some of the proposed analyses are feasible and would support the presented results and drawn conclusions. Please find my detailed comments regarding additional analyses below.

1. While I understand the authors’ argument for not doing redundancy reduction, I still believe that not performing redundancy reduction could indicate an overestimation of performance especially when comparing their method to other methods. Therefore, I believe the authors should provide additional results to ensure that the performance of CrowdGO is in fact competitive to the top CAFA3 competitors.

1.1 The authors added a new figure (Figure S3) to show the redundancy between their test and training set and claim that this distribution is similar to those for the CAFA datasets. With this figure, they provide a justification for not running redundancy reduction between their training and test set. However, how would that figure look for the CAFA targets compared to the training set? Is that similar to the CAFA3 training set and the CAFA3 targets?

1.2. The authors compare their method to the CAFA3 competitors on the CAFA3 targets. However, they use data which were not available for the CAFA3 competitors and might be similar to the CAFA3 targets, i.e., their performance could only be due to new information present in their training set that was not available for the CAFA3 competitors. How would the performance of CrowdGO look if they only use data for training available during CAFA3?

1.3 If this analysis is not possible, how would the performance of the 4 individual predictors look for the CAFA3 targets if those were trained on the training set of CrowdGO?

1.4 I follow the authors’ argument that their dataset rather presents the real-world scenario than a redundancy reduced dataset. However, using a redundancy reduced set would provide a more conservative performance estimate. I believe it would be interesting to see those results at least in the Supporting Material.

1.5 Alternatively, if re-training is too time-consuming, the authors could at least provide an assessment of the performance for different levels of sequence identity to provide some intuition on how well the method would perform for proteins similar/dissimilar to the used training set

2. Assessment of CrowdGO models using less than 4 predictors: I understand that the meta-predictor is most likely to perform best when all predictors are included, and I follow the authors’ intuition that the performance will decrease if predictors are excluded. However, why did the authors not just add a detailed ablation study? I believe it is common in the field of machine learning to assess how much different input features contribute to the final predictor. In the case of CrowdGO, I therefore still believe it would be interesting to see how a consensus predictor would perform for a subset of the 4 predictors. I don’t believe that those subsets will outperform CrowdGO and that any of those models should be provided to the users, but I would still find it interesting to see a detailed analysis of the individual contributions of the 4 predictors. This can only help users who want to build their own models to decide which predictor to include/exclude in the future.

3. How would the performance look if the 4 predictors were combined with e.g., a simple majority vote instead of training an AdaBoost? I agree with the authors that a simple majority vote would probably lead to a worse performance than the sophisticated AdaBoost. However, I still would like to see how much worse. I think without showing the actual data, their argument for performing “time-consuming” training (compared to just simply combining outputs) is rather weak while showing how other combinations would perform can only strengthen their argument for using AdaBoost. Also, in addition to just majority voting, the authors could calculate a simple average probability. In this case, high-confidence annotations from one single method will still be accepted/low-confidence annotations from many methods could still be rejected.

4. Application to full proteomes: I agree that just applying homology-based inference without any threshold will result in many false positive. However, how many of the annotations from CrowdGO could be obtained through homology-based inference using a common E-value threshold or similar? Again, I understand why applying CrowdGO is (probably) superior to applying homology-based inference. However, if the authors believe that why not support it with the respective results?

Reviewer #2: The authors have addressed by comments sufficiently.

**Have the authors made all data and (if applicable) computational code underlying the findings in their manuscript fully available?**

Reviewer #1: Yes

Reviewer #2: Yes

PLOS authors have the option to publish the peer review history of their article (what does this mean?). If published, this will include your full peer review and any attached files.

Reviewer #1: No

Reviewer #2: No

Figure Files:

Data Requirements:

Reproducibility:

References:

---

## [Editor Report · Decision Letter 2]

4 Apr 2022

Dear Dr. Waterhouse,

We are pleased to inform you that your manuscript 'CrowdGO: machine learning and semantic similarity guided consensus Gene Ontology annotation' has been provisionally accepted for publication in PLOS Computational Biology. We appreciate your attention to the reviewer's comments.

Best regards,

Jacquelyn S. Fetrow

Associate Editor

PLOS Computational Biology

Dina Schneidman

Software Editor

PLOS Computational Biology

---

## [Editor Report · Acceptance letter]

21 Apr 2022

PCOMPBIOL-D-21-01327R2 

CrowdGO: machine learning and semantic similarity guided consensus Gene Ontology annotation

Dear Dr Waterhouse,

I am pleased to inform you that your manuscript has been formally accepted for publication in PLOS Computational Biology. Your manuscript is now with our production department and you will be notified of the publication date in due course.

With kind regards,

Agnes Pap
